# Effect of Social Support on Career Decision-Making Difficulties: The Chain Mediating Roles of Psychological Capital and Career Decision-Making Self-Efficacy

**DOI:** 10.3390/bs14040318

**Published:** 2024-04-12

**Authors:** Aibao Zhou, Jintao Liu, Chunyan Xu, Mary C. Jobe

**Affiliations:** 1School of Psychology, Northwest Normal University, Lanzhou 730070, China; zhouab@nwnu.edu.cn; 2Research Center for Urban Social Psychology, School of Education, Lanzhou City University, Lanzhou 730070, China; chunyanxu68@outlook.com; 3Department of Psychological and Brain Sciences, The George Washington University, Washington, DC 20052, USA; mary.jobe.16@cnu.edu

**Keywords:** college students, social support, career decision-making difficulties, psychological capital, career decision-making self-efficacy

## Abstract

This present study explores the effect of social support on career decision-making difficulties, with the chain mediation of psychological capital and career decision-making self-efficacy. A total of 770 college students were recruited to complete the survey, which included a social support, career decision-making self-efficacy, psychological capital scale, and career decision-making difficulties scales. Significant correlations were found between social support, career decision-making difficulties, psychological capital, and career decision making self-efficacy. Path analysis indicated that the direct effect of social support on career decision-making difficulty was non-significant; social support affected career decision-making difficulties indirectly through not only the mediating effect of psychological capital but also through the chain mediation of psychological capital and career decision-making self-efficacy. Overall, the results show that social support could exert an effect on career decision-making difficulties through the mediational chains of career decision-making self-efficacy and psychological capital; the implications of this are discussed.

## 1. Introduction

Occupations play a central role in an individual’s life and can have a significant effect on one’s socioeconomic status, well-being, and the maintenance and development of individual mental health [1]. With the development of higher education in China and the implementation of an enrollment expansion policy, the employment problem of college students has become more severe and the pressure of finding employment has been constantly increasing. Exploring, choosing, and committing to a profession is a major developmental task for emerging adults [2,3]. Employment problems faced by college students, such as employment pressure and career decision-making difficulties, have become hot and tough issues of social concern. Meanwhile, career decision making is a difficult and complex process. Individuals must comprehensively consider the career itself and various factors of fit for oneself [4]. Therefore, many people will have career decision-making difficulties during this process [5]. To help college students improve their career decision-making ability, discussing the factors that affect career decision making is particularly important.

Krumboltz [6] understood that an individual’s discontent with the decision-making process was one the primary reasons for career decision-making difficulties, along with insufficient empirical learning related to the profession, and that the individual had not yet learned or applied a set of systematic methods for making career decisions. Tokar et al. [7] defined career decision-making difficulties as “the inability of an individual to choose and commit to a career choice.” Zhang et al. [8] proposed that career decision-making difficulties include all the problems individuals encounter in the early, middle, and late stages of career decision-making, which can overwhelm and impede them from accepting a conclusive career decision. Because of the professional uncertainty and shallow knowledge of career choice, an individual (at both the entry level and career transition stage) faces certain difficulties [9]. More simplistically, career decision-making difficulties can be defined as various tough problems encountered by individuals in the career decision-making process [10].

What makes college students become distracted from making career decisions has always been a focus question that researchers are trying to answer. Previously, researchers have included variables that comprised of both internal and external factors [3]. For example, the internal factors primarily entail ability, self-efficacy, psychological capital, and personality, while the external factors mainly include psychological separation, parental attachment, parental support, and social support. Social support is attributed to emotional, instructive, or functional aid from significant others (e.g., friends or family), and the support can be obtained from others or perceived by oneself [11]. Individuals need others, and others also need individuals. The close connection between individuals and others can help college students cope with various pressures in life and study [12]. High social support reduces individuals’ traumatic stress and depressive symptoms, while low social support reduces psychological resilience and psychosocial functioning [13]. In the process of career choosing, the support from parents, schools, and peers can reduce the psychological pressure reaction, alleviate mental tension, and improve positive feelings for individuals [14]. Through the emotional support, an individual receives empathy, encouragement, and a better understanding regarding the challenges attributed to the career decision making. In contrast, functional support enables guidance, information, educational pathways, tangible assistance, or any other professional opportunities [15]. Chan et al. [16] demonstrated that higher perceived social support enhances one’s effective career decision-making ability. Previous empirical studies have shown that college students’ social support was significantly positively related to their psychological capital [17,18,19]. Career decision-making self-efficacy is also closely associated with social support and has been shown to be a critical component of success for college students’ career choices [18]. Previous studies have reported finding positive associations between social support and career decision-making self-efficacy among the college students [17,20,21]. Mostly, the perceived social support boosts the college student’s career decision making through the confidence boosting trajectory [22] and ultimately impacts the career decision self-efficacy [23]. However, in terms of career decision-making difficulties, social support has also been found to be significantly correlated, though negatively [19]. For example, college students with more social support perceived less difficulty in career decision making [24]. When an individual perceives good social support, they will face less career decision-making difficulty. Thus, obtaining career information will seem less threatening and easier to approach, impacting the preparation and choice for determining whether a career should be pursued [25]. However, students should be given distinct support, and more support is not necessarily better [26].

Psychological capital is the core element of an individual’s general positivity [27] and their internal positive mental developmental state during growth process [26,28]. As an integrated mental state, psychological capital positively impacts individuals’ professional behaviors, including self-confidence (or self-efficacy), hope, optimism, and resilience [29]. The existing literature indicated the positive correlation between psychological capital and career decision-making self-efficacy [17,30]. The prior literature does indicate that psychological capital could positively predict career decision-making self-efficacy [31]. Meanwhile, psychological capital has also found to be significantly correlated with career decision-making difficulties [19,32] and positively predicted career decision-making difficulties [32].

Career decision-making self-efficacy denotes the introspection or self-reliance of a decision-maker’s abilities that are necessary to complete various tasks in career decision making [33]. For instance, the confidence levels for abilities that individuals need to complete tasks are relevant to career decision making (e.g., self-evaluation, goal selection, information collection, planning, and problem-solving). Career decision-making self-efficacy is a significant determinant of career adaptability among college students [34], and elevated self-efficacy is likely to construct professional competence [35]. Individuals with assertive career decision-making self-efficacy wield more substantial behavioral persistence with more effort, which ultimately contributes to better performance of achievement behavior [36]. It has been reported that career decision-making self-efficacy affects a range of individual choices, such as occupational identity, job search behavior, attitudes towards work, motivation to strive for performance, and efforts in occupational activities, and it is also one of the indicators of individual work performance [37,38].

Career decision-making difficulties are substantially associated with professional self-efficacy [39], and there is a significant association between them [40]. Studies have also shown that career decision-making self-efficacy mediates the association between psychological capital and college students’ career decision-making difficulties [41]. The effect of social support on college students’ career decision-making self-efficacy is indirectly made through psychological capital, which completely mediates between social support and career decision-making self-efficacy [17]. In summary, previous studies have separately explored the association between social support, psychological capital, career decision-making self-efficacy, and career decision-making difficulties. However, there is a lack of a comprehensive and systematic approach to explore how social support, career decision-making self-efficacy, and psychological capital influence career decision-making difficulties among college students. In light of the existing literature, the present study intends to use social support as the independent variable, career decision-making difficulty as the dependent variable, and career self-efficacy and psychological capital as the intermediary variables to establish a multiple intermediary model (Figure 1). We propose the following hypotheses (**H**):

**Hypothesis 1 (H1):** *Social support has a significant direct effect on career decision-making difficulties*.

**Hypothesis 2 (H2):** *Psychological capital has mediating effect on the association between social support and career decision-making difficulties*.

**Hypothesis 3 (H3):** *Career decision-making self-efficacy has a mediating effect on the association between social support and career decision-making difficulties*.

**Hypothesis 4 (H4):** *There is a chain mediating effect of psychological capital and career decision-making self-efficacy on the association between social support and career decision-making difficulties*.

In summary, this study will explore the mechanism of social support on college students’ career decision-making difficulties and examine the chain mediating role of psychological capital and career decision-making self-efficacy on this relationship. This research can provide an empirical basis for the employment guidance of college students, help them to better psychologically and socially adjust to life after college, and aims to provide theoretical references and constructive suggestions for college students’ career decision making.

## 2. Method

### 2.1. Participants

The participants were selected from three colleges and universities for each grade via the stratified cluster sampling technique. A total of 960 survey booklets were collected, and after the elimination of the incomplete responses, 770 valid survey booklets were included for further analysis; the effective response rate was 80.2%. Among them, there were 285 freshmen (37%), 199 sophomores (25.8%), and 286 juniors (37.1%); 233 men (30.3%) and 537 women (69.7%); and 572 (74.3%) who grew up in rural areas and 198 (25.7%) from urban areas. In terms of their major, 369 students (47.9%) were in professional liberal arts, 207 (26.9%) in science, and 194 (25.2%) in the arts. For plans after college, 401 students (52.1%) had working intentions and 369 (47.9%) were without working intentions; 385 students (50%) planned on taking a postgraduate entrance examination after graduation, 309 students (40.1%) were going to be directly employed, 14 students (1.8%) planned on going abroad for further studies, and 62 students (8.1%) chose something else. The demographic characteristics of the participants are shown in Table 1.

### 2.2. Measures

#### 2.2.1. Perceived Social Support Scale

This study used the revised Chinese version of the Perceived Social Support Scale compiled by Zimet et al. [42] and revised by Jiang et al. [43]. The scale has a total of 12 items that are classified into three dimensions: family support, friend support, and other people’s support. This is a 7-point Likert-type scale ranging from 1 (strongly disagree) to 7 (strongly agree). A higher total score denotes a higher degree of perceived social support by the respondents. The examples in this scale are “When I encounter difficulties, some people (leaders, relatives, classmates) will accompany me”, “I can talk about my problems with my family”, and “My friends can share happiness and sadness with me”. The internal consistency of the scale was assessed using Cronbach’s α coefficient, yielding a value of 0.94, indicating robust internal consistency and scale reliability and validity. Additionally, confirmatory factor analysis revealed favorable goodness of fit statistics: CMIN/*df* = 2.71, TLI (NNFI) = 0.96, CFI = 0.98, SRMR = 0.03, RMSEA = 0.06. 

#### 2.2.2. Positive Psychological Capital Scale

This study used the positive psychological capital scale compiled by Zhang et al. [44]. There are 26 items in the scale, which is divided into four dimensions: confidence, resilience, optimism, and hope. This is a 7-point Likert scale ranging from 1 (completely non-compliant) to 7 (completely compliant). The higher the score, the stronger the psychological capital of the respondents. The examples in this scale are “Many people appreciate my talent”, “I don’t like to be angry”, and “I will calmly seek solutions to problems in the face of difficulties”. The internal consistency of the scale was assessed using Cronbach’s α coefficient, yielding a value of 0.93, indicating robust internal consistency and scale reliability and validity. Additionally, confirmatory factor analysis revealed favorable goodness of fit statistics: CMIN/*df* = 2.93, TLI (NNFI) = 0.93, CFI = 0.94, SRMR = 0.04, RMSEA = 0.08.

#### 2.2.3. Career Decision-Making Self-Efficacy Scale

This study used the career decision-making self-efficacy scale compiled by Peng and Long [33]. There are 39 items in the scale, which is divided into five dimensions: self-evaluation, collect information, select target, planning, and problem solving. This is a 5-point Likert scale ranging from 1 (no confidence at all) to 5 (full confidence). The higher the score, the stronger the self-efficacy of career decision-making. Examples in this scale are “I can list several occupations or jobs I am interested in” and “I can make career decisions without worrying about whether they are right or wrong”. The internal consistency of the scale was assessed using Cronbach’s α coefficient, yielding a value of 0.94, indicating robust internal consistency and scale reliability and validity. Additionally, confirmatory factor analysis revealed favorable goodness of fit statistics: CMIN/*df* = 2.56, TLI (NNFI) = 0.96, CFI = 0.97, SRMR = 0.04, RMSEA = 0.06.

#### 2.2.4. Career Decision-Making Difficulties Scale

This study used the career decision-making difficulties scale for college students developed by Du and Long [45]. There are 16 items in the scale, which is divided into four dimensions: information exploration, self-exploration, planning exploration, and target anchoring. This is 5-point Likert scale that ranges from 1 (completely non-compliant) to 5 (completely compliant). The higher the score, the lower the difficulty the interviewees had in the career decision-making process. The examples in this scale are “I pay attention to the information related to my future career” and “I will learn about my personality, interests and abilities through some assessments”. The internal consistency of the scale was assessed using Cronbach’s α coefficient, yielding a value of 0.91, indicating robust internal consistency and scale reliability and validity. Additionally, confirmatory factor analysis revealed favorable goodness of fit statistics: CMIN/*df* = 3.27, TLI (NNFI) = 0.92, CFI = 0.94, SRMR = 0.05, RMSEA = 0.08. 

### 2.3. Statistical Analysis

This study used IBM SPSS 27.0 to carry out the data entry and conduct descriptive statistics and other related analyses. Additionally, Mplus 8.0 was used to test the mediating effect of psychological capital and career decision-making self-efficacy on the social support and career decision-making difficulties relationship.

## 3. Results

### 3.1. Common Method Deviation Test

Self-reporting data were used for the present study, which may lead to anticipated common method deviations. Consequently, during the study, the anonymity of the respondents was protected and the reverse scoring technique was utilized to control some entries. Harman’s single-factor method [46] was adopted for testing common method deviation to ensure the rigorousness and scientificalness of the research results. The test result shows 16 common factors with a characteristic root greater than 1, and the first factor explains 25.93% of the total variance (less than the critical standard of 40%) [47], which implies no obvious common method deviation for this study.

### 3.2. Descriptive Statistics and Correlation Analysis of the Main Variables

It was found through a correlation analysis that social support, career decision-making self-efficacy, psychological capital, and career decision-making difficulties were all significantly positively correlated (see Table 2).

### 3.3. Test of Chain Mediation Model

The structural equation model was used to analyze the chain mediating effect of psychological capital and career decision-making self-efficacy on the social support and career decision-making difficulties relationship. The results indicated that the fitting indices of the model are all acceptable (CMIN/*df* = 4.27, root mean square error of approximation = 0.07, standardized root mean square residual = 0.04, comparative fit index = 0.96, Tucker–Lewis index = 0.95 [48].

The results of the mediating effect analysis are presented in Figure 2, which indicates that social support predicted career decision-making difficulties through three paths: indirect path 1: social support → career decision-making self-efficacy → career decision-making difficulties; indirect path 2: social support → psychological capital → career decision-making difficulties; and indirect path 3: social support → psychological capital → career decision-making self-efficacy → career decision-making difficulties. First, the direct path of social support to career decision-making difficulties was non-significant (γ = 0.06, t = 0.27, *p* = 0.56), suggesting that social support might affect career decision-making difficulties indirectly. Secondly, social support positively predicted career decision-making self-efficacy (γ = 0.56, t = 2.52, *p* < 0.001) and career decision-making self-efficacy positively predicted career decision-making difficulties (γ = 0.49, t = 11.65, *p* < 0.001). This indicated that social support might indirectly affect career decision-making difficulties by career decision-making self-efficacy. Furthermore, social support positively predicted psychological capital (γ = 0.49, t = 11.89, *p* < 0.001) and psychological capital positively predicted career decision-making difficulties (γ = 0.33, t = 6.50, *p* < 0.001). These indirectly revealed that there could be a possible effect of social support on career decision-making difficulties through psychological capital. Finally, psychological capital positively predicted career decision-making self-efficacy (γ = 0.56, t = 13.74, *p* < 0.001), implying the significance of perceived social support for college students. Conclusively, elevated perceived social support from parents, friends, and others is evident in college students’ positive mental abilities and psychological capital, which denoted their strength in career decision-making self-efficacy and lessened career decision-making difficulties (see Figure 2).

The bias-corrected bootstrap was used by repeated sampling 5000 times, with deviation to test the model’s significance for the mediating effects. Table 3 shows that all three indirect effects were statistically significant since their respective bootstrap 95% confidence intervals did not contain zero. Among them, the 95% upper and lower limits of the mediating effect of career decision-making self-efficacy on the social support and career decision-making difficulties relationships were 0.01 and 0.09. Thus, the mediating effect was significant, with an effect size of 0.14. The 95% upper and lower limits for the mediating effect of psychological capital on the social support and career decision-making difficulties relationship were between 0.09 and 0.18, demonstrating it was significant; it had an effect size of 0.39. The 95% upper and lower limits for the mediating effect of psychological capital and career decision-making self-efficacy on the social support and career decision-making difficulties relationship was also found to be significant, as the limits did not contain zero (0.08, 0.15) and had an effect size of 0.33.

## 4. Discussion

Based on previous studies, this study considers social support as a predicting variable, psychological capital and career decision-making self-efficacy as intermediary variables, and career decision-making difficulties as an outcome variable, as well as establishes a chain mediation model. The study results show significantly positive correlations among social support, psychological capital, career decision-making self-efficacy, and career decision-making difficulties. However, psychological capital and career decision-making self-efficacy played chain mediating roles between social support and career decision-making difficulties. This study found that social support, psychological capital, career decision-making self-efficacy, and career decision-making difficulties were significantly positively correlated, which was consistent with previous research [17,18,19,20,21,30,34,40]. Research Hypothesis H1 was supported. Therefore, it is substantiated that social support, career decision-making self-efficacy, and psychological capital are all correlated factors that relate to career decision-making difficulties and have a positive impact on career decision-making difficulties. This is of great significance for formulating intervention plans for career decision-making difficulties among college students.

Secondly, this study uses a college student sample to explore the relationship between social support and career decision-making difficulties, with the mediating effect of psychological capital and career decision-making self-efficacy between them. Through model construction and path analysis, it was found that the direct effect between social support and college students’ career decision-making difficulties was non-significant. In contrast, the chain mediating effect of psychological capital and career decision-making self-efficacy was significant. The results show that social support positively predicted psychological capital, which is consistent with the findings of He and Yao [18]. This implies that the more social support an individual perceives, the easier it will be for them to build and enhance positive psychological capital. Additionally, psychological capital positively predicted career decision-making self-efficacy, implying the college students’ higher positive psychological resources and richer psychological capital led to an elevated career decision-making self-efficacy. Prior research is supportive of this finding, demonstrating that career decision-making difficulties are positively determined by psychological capital [17,31], signifying that individuals with more psychological capital will have an easier time making career decisions. It has also been shown that individuals with high psychological capital can better maintain their composure and discover reliable sources of information while under stress or dealing with career decision-making difficulties, allowing them to manage the different challenges of the process [32,49]. Lastly, career decision-making self-efficacy positively predicted career decision-making difficulties, which indicates that the higher career decision-making self-efficacy individuals have, the fewer career decision-making difficulties they face. This result is also consistent with previous research [50].

The mediation analysis showed that social support indirectly affected career decision-making difficulties through psychological capital and career decision-making self-efficacy. Furthermore, psychological capital and career decision-making self-efficacy played a chain-like mediating role between social support and career decision-making difficulties, supporting Hypothesis H3. Due to the mediation effect of psychological capital and career decision-making self-efficacy, the direct effect of social support on career decision-making difficulties was non-significant; therefore, the research Hypothesis H2 was not supported. Additionally, the effect of social support on career decision-making self-efficacy was mediated by the psychological capital, denoting the indirect effect of social support on career decision-making self-efficacy by influencing the psychological capital. Consistent with the previous studies [17,20], psychological capital enacted a complete mediation between social support and career decision-making self-efficacy for the present study. Furthermore, psychological capital mediated the effect of social support on career decision-making difficulties, and this path had the greatest effect size. Career decision-making self-efficacy also mediated both the impact of social support as well as the psychological capital, separately, on career decision-making difficulties. Finally, the present study explores the mediating role of career decision-making self-efficacy and psychological capital in the influence of social support on career decision-making difficulties through a path analysis. The results support that social support influences career decision-making difficulties through the chain mediation effect of career decision-making self-efficacy and psychological capital. The result specifies social support → psychological capital → career decision-making self-efficacy → career decision-making difficulties; therefore, path 4 in the hypothetical model was supported. This specific finding implies college students with higher levels of social support can feel more support from family, friends, and others in the face of adversity and thus will obtain an increased level of positive psychological capital and psychological aptitude to handle obstacles and be better able to maintain firm beliefs and make sound decisions. This is beneficial for them to boost their career decision-making self-efficacy and ultimately be able to make career decision more easily.

College students are a good population to study for this, as they are in the stage of transforming from “students” to “professionals.” When they make a career decision for the first time, they are prone to career decision-making difficulties [8]. Through the study model, social support influences career decision-making difficulties through the mediating effect of psychological capital and career decision-making self-efficacy, with the greatest mediating effect being through psychological capital. It is evident among college students that psychological capital is a key factor that affects levels of career decision-making difficulties. Positive psychological resources promote positive individual behavior, and students with higher levels of psychological capital tend to be more positive about understanding themselves and their environment. Such individuals never contradict their aptitude during adversity; rather, they actively search for solutions, which can lower the degree of difficulty in career decision making. Therefore, colleges should make it a priority, from a positive psychological standpoint, to treat students positively, bolster their students’ self-confidence, and grow their psychological capital to ultimately lessen their career decision-making difficulty. Students with a greater level of psychological capital are better prepared for the workforce because they are more likely to be proactive in their pursuit of knowledge, confident in their career decisions, organized in their approach, and realistic in their self-evaluations. Additionally, they have a deeper appreciation for their own strengths and shortcomings. Overall, these students are more equipped to make an informed decision when confronted with a career difficulty, which is an inevitable experience that is faced at the end of college.

## 5. Research Limitations and Future Prospects

The present study has some limitations that need to be improved in future research. First, the study uses self-reported data where the participants are vulnerable to the impact of social expectations leading to possible errors. Apart from that, the participants might have drawbacks due to their limited knowledge and understanding regarding some specific terms in the questionnaire (i.e., career decision-making self-efficacy and career decision-making difficulties). Some participants, especially those in lower academic years, may not have a deep understanding of their career development and career planning, which could also affect the credibility of the data. In the future, we can increase the clarity of college students’ self-evaluation of career decision making and the accuracy of the received data in a simulated environment using career decision-making scenarios. Second, this study has combined and analyzed the college student groups without considering the influence of different genders, grades, and professional categories. Undoubtedly, those demographic variables may have an unexpected impact on the research results. Future research can subdivide and compare groups while considering the differences between different genders, grades, and professional categories to better reflect these groups. Finally, although the relationship between social support (including from family, friends, and others) and college students’ career decision-making difficulties and its influence mechanism (the sequential mediation of psychological capital and career decision-making self-efficacy) have been supported, it is still necessary to conduct a longitudinal study on the causal relationship between variables, which is conducive to providing more convincing evidence to confirm the change logic of these variables in a time series. Future research can also explore more environmental variables to examine various factors affecting college students’ career decision making.

## 6. Conclusions

In light of the findings, the present study signifies the correlations among college students’ social support, psychological capital, career decision-making self-efficacy, and career decision-making difficulties. The findings imply that social support positively predicts psychological capital and psychological capital positively predicts career decision-making self-efficacy and career decision-making difficulties. Likewise, career decision making self-efficacy positively predicts career decision-making difficulties. We found that psychological capital significantly mediates the relationship between social support and career decision-making self-efficacy; also, the intermediary effect of career decision-making self-efficacy between psychological capital and career decision-making difficulties is significant. The present study signifies positive psychological resources help an individual alleviate career decision-making difficulties and promote career decision-making self-efficacy. Above all, higher social support from family, friends, and significant others helps college students gain positive psychological capital and boosts their career decision-making self-efficacy to confront career decision-making difficulties spontaneously and more easily. For implementing the research insights, social support framework and policy can be fostered with mindfulness, positive psychology practices, and resilience training for students who are embarking on a fresh professional journey. Moreover, providing personalized career counseling can enhance their career decision-making self-efficacy for informed and improved career choices.

## Figures and Tables

**Figure 1 behavsci-14-00318-f001:**
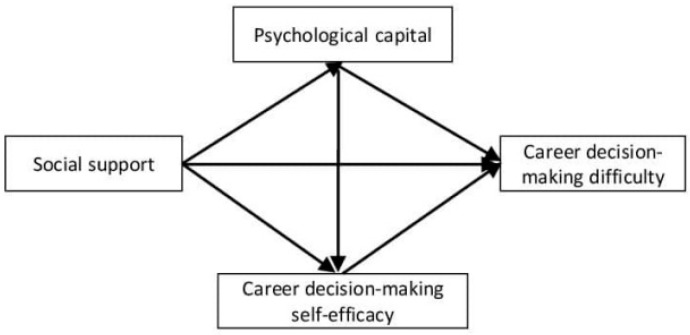
Hypothetical chain mediation model.

**Figure 2 behavsci-14-00318-f002:**
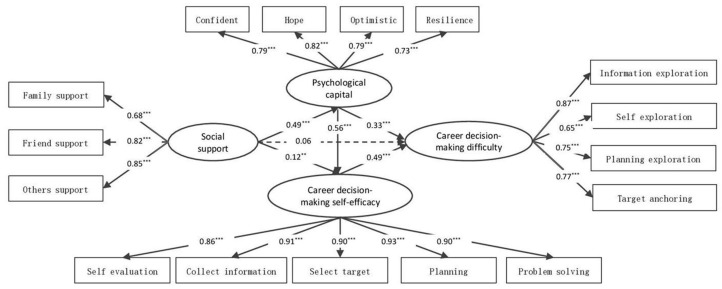
The chain mediating effect model of psychological capital and career decision-making self-efficacy between social support and career decision-making difficulties. *Note*. ** *p* < 0.01, *** *p* < 0.001.

**Table 1 behavsci-14-00318-t001:** Demographic information.

Variables	Groups	Frequency (%)
Gender	Woman	233 (30.3%)
	Man	537 (69.7%)
Grade	Freshmen	285 (37%)
	Sophomores	199 (25.8%)
	Juniors	286 (37.1%)
Residence	Urban	198 (25.7%)
	Rural	572 (74.3%)
Ethnic Group	Han Chinese	672 (87.3%)
	Minorities	98 (12.7%)
Professional Category	Liberal Arts	369 (47.9%)
	Science	207 (26.9%)
	Arts	194 (25.2%)

**Table 2 behavsci-14-00318-t002:** Descriptive statistics and correlation coefficients of various variables.

Variables	*M* ± *SD*	1	2	3	4
1. Social Support	5.07 ± 0.91	1			
2. Career Decision-Making Self-Efficacy	3.18 ± 0.62	0.365 **	1		
3. Psychological Capital	4.67 ± 0.71	0.417 **	0.527 **	1	
4. Career Decision-Making Difficulties	3.39 ± 0.59	0.364 **	0.645 **	0.526 **	1

*Note*. *M* = mean; *SD* = standard deviation. ** *p* < 0.01.

**Table 3 behavsci-14-00318-t003:** Bootstrap analysis and effect size of the significance test of the mediation effect.

Path	Standardized Indirect Effect Estimates	Effect Rate	95% Confidence Interval	
			Lower Limit	Upper Limit
Social support → career decision-making difficulties	0.06	14%	−0.03	0.11
Social support → career decision-making self-efficacy → career decision-making difficulties	0.06	14%	0.01	0.09
Social support → psychological capital → career decision-making difficulties	0.16	39%	0.09	0.18
Social support → psychological capital → career decision-making self-efficacy → career decision-making difficulties	0.14	33%	0.08	0.15
Direct effect	0.06	14%		
Indirect effect	0.36	86%		
Total effect	0.42	100%		

## Data Availability

The research data will be available upon request to the corresponding author.

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
