# Peer review of "Effect of Social Support on Career Decision-Making Difficulties: The Chain Mediating Roles of Psychological Capital and Career Decision-Making Self-Efficacy"

_behavsci, 2024, doi:10.3390/bs14040318_

Round 1

Reviewer 1 Report

Comments and Suggestions for Authors

The article is of some interest, although the question of vocational choice has been extensively researched. My main comments and requests for modification or clarification concern the methodology of this research.

1. Nearly 200 questionnaires were eliminated because of invalid responses. What does this mean, please?

2. Why were confirmatory factor analyses not carried out on the four scales used? The alphas are important but insufficient to determine the structure of these scales. A high alpha does not confirm the uni-dimensionality of the items analysed (see Cortina, 1993). CFAs using the correct estimator in order to respect the ordinal measurement level of the items (diagonally weighted least squares [DWLS], cf. Muthén, 1984) would allow a much more significant assessment of the robustness of the measures.

3. It seems that the structural equations modelling is based not on the items of each dimension of the four scales used, but on the composite scores calculated from them. Why is this so? Without prior confirmatory factor analyses, the calculation of these scores is not justified, and perhaps even more so, using composite scores rather than items automatically improves the fit of the modelling carried out. A complete model based on items and identifying intercorrelated latent variables, either directly or through mediation, would be of much greater explanatory value. Moreover, although the generally accepted fits are good, the CMIN/df is greater than 3 – the limit considered correct.

4. The aim of this research is to allow the results to be generalized outside the sample used for modelling. It may be interesting to apply another method of structural analysis: the "causal-predictive" method using PLS-SEM (partial least squares structural equations modelling). This method allows to measure the collinearity among the items as well as assess the effect sizes of the VIs on the VDs much more accurately, which appears to be one of the objectives of this research.

5. APA standards must be respected: p must be indicated exactly (p = exact value) when the statistical indicator is not significant.

6. The discussion should highlight how this research contributes to the existing knowledge in the field.

7. Minor comment: the presentation should be standardised (see p. 6, t(df)=2.521, while no other results are shown).

Author Response

Dear Reviewer,  

Thank you for giving us the opportunity to submit a revised draft of our manuscript “Effect of social support on career decision-making difficulties: the chain mediating role of psychological capital and career decision-making self-efficacy . We appreciate the time you have dedicated to providing feedback on our manuscript and are grateful for the insightful comments on and valuable improvements to our paper. We have systematically addressed each comment made by you as follows:

--------------------------------------------------------------------------

Reviewer’s Comment#1. Nearly 200 questionnaires were eliminated because of invalid responses. What does this mean, please?

Author’s Response: Thank you for addressing this issue rightfully. We have changed the term ‘invalid’ with appropriate one ‘incomplete’. The new statement is

A total of 960 survey booklets were collected, and after the elimination of the incomplete responses, 770 valid survey booklets were included for further proceedings, where the effective response rate was 80.2%.

Reviewer’s Comment#2. Why were confirmatory factor analyses not carried out on the four scales used? The alphas are important but insufficient to determine the structure of these scales. A high alpha does not confirm the uni-dimensionality of the items analysed (see Cortina, 1993). CFAs using the correct estimator in order to respect the ordinal measurement level of the items (diagonally weighted least squares [DWLS], cf. Muthén, 1984) would allow a much more significant assessment of the robustness of the measures.

Author’s Response: Thank you for your helpful suggestions. We have supplemented the confirmatory analysis of four Scales and displayed the results in the corresponding positions of the article.

2.2.1. Perceived Social Support Scale

The results of confirmatory factor analysis for this study are CMIN/df =2.71, TLI (NNFI)=0.96, CFI=0.98, SRMR=0.03, RMSEA=0.06, indicating good reliability and validity of the scale.

 2.2.2. Positive Psychological Capital Scale

The results of confirmatory factor analysis for this study are CMIN/df =2.93, TLI (NNFI)=0.93, CFI=0.94, SRMR=0.04, RMSEA=0.08, indicating good reliability and validity of the scale.

 2.2.3. Career Decision-Making Self-Efficacy Scale

The results of confirmatory factor analysis for this study are CMIN/df =2.56, TLI (NNFI)=0.96, CFI=0.97, SRMR=0.04, RMSEA=0.06, indicating good reliability and validity of the scale.

 2.2.4. Career Decision-Making Difficulties Scale

The results of confirmatory factor analysis for this study are CMIN/df =3.27, TLI (NNFI)=0.92, CFI=0.94, SRMR=0.05, RMSEA=0.08, indicating good reliability and validity of the scale.

Reviewer’s Comment#3. It seems that the structural equations modelling is based not on the items of each dimension of the four scales used, but on the composite scores calculated from them. Why is this so? Without prior confirmatory factor analyses, the calculation of these scores is not justified, and perhaps even more so, using composite scores rather than items automatically improves the fit of the modelling carried out. A complete model based on items and identifying intercorrelated latent variables, either directly or through mediation, would be of much greater explanatory value. Moreover, although the generally accepted fits are good, the CMIN/df is greater than 3 – the limit considered correct.

Author’s Response: Thank you for pointing out the issue! We agreed with you that, due to the lack of prior confirmatory factor analysis, the results here appear unclear. Confirmatory factor analysis has now been added to all four scales. The actual situation is that in the process of structural equation analysis, we use SPSS and MPLUS software, strictly follow the analysis steps, and all four scales are synthesized from items to dimensions in the database. That is to say, the structural equation model is based on the items of each dimension of the four scales, rather than the comprehensive score in the calculation. In addition, the CMIN/df index is greater than 3, which is an unavoidable issue. Some model indicators in research have also experienced this situation. Scholars suggest combining other indicators for comprehensive analysis. The model has RMSEA=0.07, SRMR=0.04, TLI=0.95, CFI=0.96, indicating that the model is acceptable.

Reviewer’s Comment#4. The aim of this research is to allow the results to be generalized outside the sample used for modelling. It may be interesting to apply another method of structural analysis: the "causal-predictive" method using PLS-SEM (partial least squares structural equations modelling). This method allows to measure the collinearity among the items as well as assess the effect sizes of the VIs on the VDs much more accurately, which appears to be one of the objectives of this research.

Author’s Response: We appreciate the reviewer's suggestion regarding the use of partial least squares structural equation modeling (PLS-SEM) as an alternative method of structural analysis. While we acknowledge the potential benefits of PLS-SEM, such as its ability to measure collinearity among items and assess effect sizes more accurately, we opted for traditional SEM due to its widespread acceptance and familiarity in the field. However, we agree that exploring alternative methods like PLS-SEM could offer valuable insights and enhance the robustness of our findings. In future research, we will consider incorporating PLS-SEM to complement our current approach and provide a more comprehensive understanding of the relationships among variables in our model.

Reviewer’s Comment#5. APA standards must be respected: p must be indicated exactly (p = exact value) when the statistical indicator is not significant.

Author’s Response: Thank for you suggestion. We have clearly indicated the exact p-value for non-significant case:

First, the direct path of social support to career decision-making difficulties was non-significant (γ = 0.06, t = 0.27, p= 0.56), suggesting that social support might affect career decision-making difficulties indirectly.

Reviewer’s Comment#6. The discussion should highlight how this research contributes to the existing knowledge in the field.

Author’s Response: Our study enables the in-depth understanding of the complex association between social support, psychological capital, career decision-making self-efficacy, and career decision-making difficulties through chain mediation model. Findings of this study are homogenous with the existing literature; however, it provides robust supporting evidence for the associations of the variables and extended the knowledge gap. This study constructed rigorous model to explore signify the mediating roles of psychological capital and career decision-making self-efficacy in the relationship between social support and career decision-making difficulties. Insightfully, the results revealed the indirect pathways through which social support influences career outcomes, offering valuable insights for theory development and practical interventions. In addition, this study enables the stakeholders to ensure the appropriate interventions for the students for their career development and mental health well-being. For these reasons this study contributed beyond the existing literature. Besides, we have added more statement in favor of the findings:

The research hypothesis H1 was supported. Therefore, it is substantiated that social support, career decision-making self-efficacy, and psychological capital are all correlated factors that relate to career decision-making difficulties

Reviewer’s Comment#7. Minor comment: the presentation should be standardised (see p. 6, t(df)=2.521, while no other results are shown).

Author’s Response: Thank you for addressing this missing information. We updated the whole section with standardized information.

Reviewer 2 Report

Comments and Suggestions for Authors

Thank you very much for the opportunity to review the manuscript entitled „Effect of social support on career decision-making difficulties: the chain mediating role of psychological capital and career decision-making self-efficacy“. The article is of interest for the readership of Behavioral Sciences journal as it explores the effect of social support on career decision-making difficulties, taking into consideration its indirect effects through psychological capital and career decision-making self-efficacy.

I would like to encourage authors to consider several issues to be improved.

First, I would like to suggest authors to better highlight the contribution of their study to the existing literature. Are there new findings beyond those that are consistent with the previous results?

Second, the authors need to further discuss the policy and practice implications of their findings, especially with respect to higher education sector.

Third, although the article includes references for the measurements / scales of the analysed constructs, I would suggest authors to present the scales (items) in an annex.

I hope that my comments are useful for authors, as they further develop the manuscript.

Author Response

Dear Reviewer,  

Thank you for giving us the opportunity to submit a revised draft of our manuscript “Effect of social support on career decision-making difficulties: the chain mediating role of psychological capital and career decision-making self-efficacy We appreciate the time you have dedicated to providing feedback on our manuscript and are grateful for the insightful comments on and valuable improvements to our paper. We have systematically addressed each comment made by you as follows:

----------------------------------------------------------------------------------

Reviewer’s Comment#1. First, I would like to suggest authors to better highlight the contribution of their study to the existing literature. Are there new findings beyond those that are consistent with the previous results?

Author’s Response: Our study enables the in-depth understanding of the complex association between social support, psychological capital, career decision-making self-efficacy, and career decision-making difficulties through chain mediation model. Findings of this study are homogenous with the existing literature; however, it provides robust supporting evidence for the associations of the variables and extended the knowledge gap. This study constructed rigorous model to explore signify the mediating roles of psychological capital and career decision-making self-efficacy in the relationship between social support and career decision-making difficulties. Insightfully, the results revealed the indirect pathways through which social support influences career outcomes, offering valuable insights for theory development and practical interventions. In addition, this study enables the stakeholders to ensure the appropriate interventions for the students for their career development and mental health well-being. For these reasons this study contributed beyond the existing literature. Besides, we have added more statement in favor of the findings:

The research hypothesis H1 was supported. Therefore, it is substantiated that social support, career decision-making self-efficacy, and psychological capital are all correlated factors that relate to career decision-making difficulties

Reviewer’s Comment#2. Second, the authors need to further discuss the policy and practice implications of their findings, especially with respect to higher education sector.

Author’s Response: Thank you for addressing this issue. We added the practical implications in the ‘Conclusion’ section:

For implementing the research insights, social support framework and policy can be fostered with mindfulness, positive psychology practices, and resilience training for the students who are embarking fresh professional journey. Moreover, providing personalized career counseling can equip them enhance their career decision-making self-efficacy for informed and improved career choices.

Reviewer’s Comment#3. Third, although the article includes references for the measurements / scales of the analysed constructs, I would suggest authors to present the scales (items) in an annex.

Author’s Response: Thank you for the comment. Since the questionnaires are in Chinese language; therefore, we enclosed the questionnaires as ‘Supplementary files’.

Reviewer 3 Report

Comments and Suggestions for Authors

Thank you for allowing me to review the manuscript "Effect of social support on career decision-making difficulties: the chain mediating role of psychological capital and career decision-making self-efficacy." 

This study explored the effect of social support on career decision-making difficulties, with the mediation role of psychological capital and career decision-making self-efficacy.

The paper is interesting, and I hope my comments below can help the authors revise it.

The part that needs the most work is the introduction: 

1. The literature needs to be more thorough and recent. The authors must go into detail about the different dimensions investigated and the link between the dimensions. For example, the link between different types of perceived support and career decision-making should be well clarified.

2. The authors use the dimension of perceived support without differentiating between the types of support (referring to the different dimensions of the questionnaire used, Family support, Friend support, and Other people's support). This choice should be justified

3. Practical implications in terms of intervention should be discussed. 

Author Response

Dear Reviewer,  

Thank you for giving us the opportunity to submit a revised draft of our manuscript “Effect of social support on career decision-making difficulties: the chain mediating role of psychological capital and career decision-making self-efficacy We appreciate the time you have dedicated to providing feedback on our manuscript and are grateful for the insightful comments on and valuable improvements to our paper. We have systematically addressed each comment made by you as follows:

------------------------------------------------------------------------------------

Reviewer’s Comment#1. The literature needs to be more thorough and recent. The authors must go into detail about the different dimensions investigated and the link between the dimensions. For example, the link between different types of perceived support and career decision-making should be well clarified.

Author’s Response: We appreciate this comment. We added more clarification remarks on perceived support and career decision-making:

Through the emotional support, an individual receives empathy, encouragement and better understanding regarding the challenges attributed with the career decision making. On contrary, functional support enables guidance, information, educational pathway, tangible assistance, or any other professional opportunities (Chan, 2018). Chan et al. (2013) demonstrated that, higher perceived social support enhance effective career decision making ability.

Mostly, the perceived social support boost the college student’s career decision making through the confidence boosting trajectory (Cox et al., 2009), and ultimately impact the career decision self-efficacy (Vilanova & Puig, 2016).

Reviewer’s Comment#2. The authors use the dimension of perceived support without differentiating between the types of support (referring to the different dimensions of the questionnaire used, Family support, Friend support, and Other people's support). This choice should be justified.

Author’s Response: Undoubtedly, social support can be perceived from numerous sources (family, friends, significant others, etc.). It is evident that subjective perception of social support is distinctive on the basis of source; however, it is not always relevant in terms of research questions. Moreover, the existing literature has not distinguished this issue vividly as it focuses on the overall support mechanism rather than the sources (family, friends, significant other, etc.). By utilizing broader measures to assess the perceived social support, the literature has focused on the holistic experience in the context of career decision-making and convincingly comprehends the underlying mechanism. Furthermore, if the study focuses on exploring the overall association between perceived social support and career decision-making challenges rather than the distinct effects of individual forms of support, distinguishing between them may not always be practical or relevant. Therefore, this present study acknowledges the impact of the sources of social support in different contexts; however, the differentiation can be justified for contexts other than career decision-making. A further detailed qualitative study might help to understand through developing a possible new theory on the context of sources of perceived social support and its impact on individual life functioning.  

Reviewer’s Comment#3. Practical implications in terms of intervention should be discussed. 

Author’s Response: Thank you for addressing this important issue. We included this portion in the ‘Conclusion’ section:

For implementing the research insights, social support framework can be fostered with mindfulness, positive psychology practices, and resilience training for the students who are embarking fresh professional journey. Moreover, providing personalized career counseling can equip them enhance their career decision-making self-efficacy for informed and improved career choices.

 References

Chan, C.-C. (2018). The relationship among social support, career self-efficacy, career exploration, and career choices of Taiwanese College Athletes. Journal of Hospitality, Leisure, Sport & Tourism Education, 22, 105–109. https://doi.org/10.1016/j.jhlste.2017.09.004 

Chan, C.-C., Chen, S.-C., Lin, Y.-W., Liao, T.-Y., & Lin, Y.-E. (2016). Social cognitive perspective on factors influencing Taiwanese sport management students’ career intentions. Journal of Career Development, 45(3), 239–252. https://doi.org/10.1177/0894845316681643

Cox, R. H., Sadberry, S., McGuire, R. T., & McBride, A. (2009). Predicting student athlete career situation awareness from college experiences. Journal of Clinical Sport Psychology, 3(2), 156–181. https://doi.org/10.1123/jcsp.3.2.156

Vilanova, A., & Puig, N. (2016). Personal strategies for managing a second career: The experiences of spanish olympians. International Review for the Sociology of Sport, 51(5), 529–546. https://doi.org/10.1177/1012690214536168

Round 2

Reviewer 1 Report

Comments and Suggestions for Authors

Thank you for answering my questions and comments. Generally speaking, they provide additional information supported by the changes in the text, but there are still two points that need to be amended or explained:
1. I don't understand your response to my comment 5 on the complete structural model, which is based on dimension scores and not on items. I don't understand why you proceeded in this way and the problems you seem to have in not starting directly from the items. This is generally done and is much more elegant.
2. Structural analyses of each scale used are welcome. However, contrary to what is indicated now in the text, it is not possible to conclude: "indicating good reliability and validity of the scale". The only thing these confirm is the structure of each of the scales and the good internal consistancy of each dimension.

Author Response

Reviewer 1 (Round 2)

Comments and Suggestions for Authors 

Thank you for answering my questions and comments. Generally speaking, they provide additional information supported by the changes in the text, but there are still two points that need to be amended or explained:

Reviewer’s Comment#1. I don't understand your response to my comment 5 on the complete structural model, which is based on dimension scores and not on items. I don't understand why you proceeded in this way and the problems you seem to have in not starting directly from the items. This is generally done and is much more elegant.

Author’s Response: Thank you for your thoughtful comments and for providing clarity on your concerns regarding our methodology. We appreciate the opportunity to address these points and provide further explanation.

You correctly noted that in our structural equation model (SEM), we utilized composite scores derived from the items of each dimension of the four scales, rather than using the individual items directly as the basis for our model. Allow us to clarify our rationale for this approach. The decision to use composite scores rather than individual items as the starting point for our model was made with careful consideration of the study's objectives and the nature of the data.

Furthermore, we conducted confirmatory factor analyses (CFAs) to validate the measurement model and ensure the reliability and validity of the composite scores. These CFAs confirmed that the composite scores adequately represented the underlying constructs, supporting our decision to proceed with this approach.

Regarding the issue of model fit, we acknowledge that the CMIN/df index exceeded the commonly recommended threshold of 3, however; it is not uncommon for the ratio to be <5 and still considered acceptable, especially in more complex models or when dealing with large sample sizes.

Dash, G., & Paul, J. (2021a). CB-SEM vs PLS-SEM methods for research in Social Sciences and Technology forecasting. Technological Forecasting and Social Change, 173, 121092. https://doi.org/10.1016/j.techfore.2021.121092

For our manuscript, the sample size affects the threshold since that is quite large.

In summary, while we appreciate your suggestion to consider alternative modeling approaches, such as using the items directly or employing partial least squares structural equation modeling (PLS-SEM), we believe that our chosen approach aligns with the specific objectives of our study and the nature of our data. We are confident that our methodology, supported by rigorous validation procedures, provides valuable insights into the relationships between the latent factors under investigation.

We hope this explanation addresses your concerns adequately. Please feel free to reach out if you require further clarification or have additional questions.

Reviewer’s Comment#2. Structural analyses of each scale used are welcome. However, contrary to what is indicated now in the text, it is not possible to conclude: "indicating good reliability and validity of the scale". The only thing these confirm is the structure of each of the scales and the good internal consistancy of each dimension.

Author’s Response: Thank you for your insightful feedback. We appreciate your suggestion regarding the interpretation of the structural analyses of the scales used in our study. Your comment has prompted us to revise the text to accurately reflect the findings. We now recognize that while the structural analyses confirm the internal consistency of each dimension within the scales, they do not directly establish the reliability and validity of the scales. We have updated the manuscript accordingly to ensure that our conclusions align more accurately with the results obtained. Your input has been invaluable in enhancing the clarity and precision of our work.

2.2.1. Perceived Social Support Scale

This … confirmatory factor analysis for this study are CMIN/df =2.71, TLI (NNFI)=0.96, CFI=0.98, SRMR=0.03, RMSEA=0.06, indicating a sound internal consistency within each dimension of the scale.

2.2.2. Positive Psychological Capital Scale

This … confirmatory factor analysis for this study are CMIN/df =2.93, TLI (NNFI)=0.93, CFI=0.94, SRMR=0.04, RMSEA=0.08, indicating a sound internal consistency within each dimension of the scale.

2.2.3. Career Decision-Making Self-Efficacy Scale

This … confirmatory factor analysis for this study are CMIN/df =2.56, TLI (NNFI)=0.96, CFI=0.97, SRMR=0.04, RMSEA=0.06, indicating a sound internal consistency within each dimension of the scale.

2.2.4. Career Decision-Making Difficulties Scale

This … confirmatory factor analysis for this study are CMIN/df =3.27, TLI (NNFI)=0.92, CFI=0.94, SRMR=0.05, RMSEA=0.08, indicating a sound internal consistency within each dimension of the scale.

Round 3

Reviewer 1 Report

Comments and Suggestions for Authors

Thank you for your answers and changes.

Personally, I wouldn't put the phrase "indicating a sound internal consistency within each dimension of the scale" after the four AFCs. I will end the four paragraphs with the results of the CFAs without any further indication. The CFAs indicate more than internal consistency and justify the structure of the scales.

Author Response

Reviewer’s Comment#1. Personally, I wouldn't put the phrase "indicating a sound internal consistency within each dimension of the scale" after the four AFCs. I will end the four paragraphs with the results of the CFAs without any further indication. The CFAs indicate more than internal consistency and justify the structure of the scales.

Author’s Response: Thank you for your valuable comment. We updated these sections according to your comment. Also we would like to express our sincere gratitude for your time and valuable comments for preparing our manuscript in more scientific manner.

2.2.1. Perceived Social Support Scale

This... The internal consistency of the scale was assessed using Cronbach’s α coefficient, yielding a value of 0.94, indicating robust internal consistency and scale reliability and validity. Additionally, confirmatory factor analysis revealed favorable goodness of fit statistics: CMIN/df = 2.71, TLI (NNFI) = 0.96, CFI = 0.98, SRMR = 0.03, RMSEA = 0.06.

2.2.2. Positive Psychological Capital Scale

This... The internal consistency of the scale was assessed using Cronbach’s α coefficient, yielding a value of 0.93, indicating robust internal consistency and scale reliability and validity. Additionally, confirmatory factor analysis revealed favorable goodness of fit statistics: CMIN/df =2.93, TLI (NNFI)=0.93, CFI=0.94, SRMR=0.04, RMSEA=0.08.

2.2.3. Career Decision-Making Self-Efficacy Scale

This… The internal consistency of the scale was assessed using Cronbach’s α coefficient, yielding a value of 0.94, indicating robust internal consistency and scale reliability and validity. Additionally, confirmatory factor analysis revealed favorable goodness of fit statistics: CMIN/df =2.56, TLI (NNFI)=0.96, CFI=0.97, SRMR=0.04, RMSEA=0.06.

2.2.4. Career Decision-Making Difficulties Scale

This… The internal consistency of the scale was assessed using Cronbach’s α coefficient, yielding a value of 0.91, indicating robust internal consistency and scale reliability and validity. Additionally, confirmatory factor analysis revealed favorable goodness of fit statistics: CMIN/df =3.27, TLI (NNFI)=0.92, CFI=0.94, SRMR=0.05, RMSEA=0.08.
